# Choline Plus Working Memory Training Improves Prenatal Alcohol-Induced Deficits in Cognitive Flexibility and Functional Connectivity in Adulthood in Rats

**DOI:** 10.3390/nu12113513

**Published:** 2020-11-14

**Authors:** Jaylyn Waddell, Elizabeth Hill, Shiyu Tang, Li Jiang, Su Xu, Sandra M. Mooney

**Affiliations:** 1Division of Neonatology, Department of Pediatrics, University of Maryland School of Medicine, Baltimore, MD 21201, USA; ehill@som.umaryland.edu (E.H.); Sandra_mooney@unc.edu (S.M.M.); 2Department of Diagnostic Radiology and Nuclear Medicine, University of Maryland School of Medicine, Baltimore, MD 21201, USA; sytang@som.umaryland.edu (S.T.); lijiang@som.umaryland.edu (L.J.); sxu@umm.edu (S.X.); 3Department of Nutrition, Nutrition Research Institute, University of North Carolina at Chapel Hill, Kannapolis, NC 28081, USA

**Keywords:** prenatal alcohol, fMRI, reversal learning, delayed non-matching to place, attentional set shifting, striatum, prefrontal

## Abstract

Fetal alcohol spectrum disorder (FASD) is the leading known cause of intellectual disability, and may manifest as deficits in cognitive function, including working memory. Working memory capacity and accuracy increases during adolescence when neurons in the prefrontal cortex undergo refinement. Rats exposed to low doses of ethanol prenatally show deficits in working memory during adolescence, and in cognitive flexibility in young adulthood. The cholinergic system plays a crucial role in learning and memory processes. Here we report that the combination of choline and training on a working memory task during adolescence significantly improved cognitive flexibility (performance on an attentional set shifting task) in young adulthood: 92% of all females and 81% of control males formed an attentional set, but only 36% of ethanol-exposed males did. Resting state functional magnetic resonance imaging showed that functional connectivity among brain regions was different between the sexes, and was altered by prenatal ethanol exposure and by choline + training. Connectivity, particularly between prefrontal cortex and striatum, was also different in males that formed a set compared with those that did not. Together, these findings indicate that prenatal exposure to low doses of ethanol has persistent effects on brain functional connectivity and behavior, that these effects are sex-dependent, and that an adolescent intervention could mitigate some of the effects of prenatal ethanol exposure.

## 1. Introduction

Prenatal alcohol exposure induces a spectrum of anatomical, behavioral, and cognitive deficits described as fetal alcohol spectrum disorder (FASD). The cognitive deficits can be generally described as poor executive function—higher-order cognition that requires efficient and flexible interaction between attention, working memory, response inhibition, and impulse control [1,2,3]. Poor planning, decision-making, and behavioral inflexibility are well characterized in children exposed to alcohol during gestation, and the magnitude of these deficits correlate with parental ratings of behavioral problems [4]. Despite the known harm prenatal alcohol exposure can cause, incidence of FASD remains stable, as an estimated 2%–5% of US children may be eligible for a diagnosis [5,6].

The essential nutrient choline was first demonstrated to improve outcomes in a rodent model of FASD, improving cognitive function and normalizing the expression of developmental milestones such as eye opening [7]. Choline plays multiple roles in brain development. It is a critical component of lipid membranes, influences DNA and histone methylation, and is a precursor of acetylcholine, a neurotransmitter that is critically involved in cognition [8,9]. Perinatal choline supplementation can improve working memory, impulsive responding, interval timing, and complex problem solving in typically developing rats [10,11,12]. These changes persist into adulthood, and the timing of effective supplementation is protracted [10,11,12]. In children with known prenatal alcohol exposure, choline supplementation between 2.5 and 5 years of age improved performance on a hippocampal-dependent cognitive task [13]. Supplementation after 5 years of age did not appear effective with 6 weeks of treatment [14]. It is possible that effective treatments in older children might require combinations of interventions that selectively target known affected behaviors and neural substrates [15,16]. Working memory and attention training as well as self-regulation training have demonstrated some efficacy in improving social cognition and verbal fluency in children with FASD [15,16].

In healthy children and adolescents, discrete subregions of the frontal cortices are selectively activated by specific task demands in tests of domains necessary for efficient executive function [17,18]. In children with FASD, however, a less-selective, general pattern of activation is evident [17]. Poor frontal lobe organization and connectivity is emerging as a paramount feature of FASD [19,20]. In rodents, this loss of adaptive frontal lobe modulation of learned responses can manifest as poor reversal learning and use of inefficient problem-solving strategies [21]. When contingencies change and require “updating”, inefficient and perseverative responding is often observed in rats exposed to alcohol (ethanol: EtOH) gestationally [19,21]. Adolescence is a time during which working memory capacity and efficiency reach adult levels [18]. As the frontal cortices mature, the ability to hold items in mind and adapt to changing contingencies becomes more efficient [22,23,24]. We previously reported improved performance on the attentional set shifting and reversal learning task (ASST-R) in rats exposed prenatally to EtOH when both postnatal choline supplementation and working memory training were combined in the juvenile phase [21]. Males exhibited more general impairments in ASST-R than females, but both sexes did exhibit more errors and higher trials to criterion in some phases of the task [21]. Efficient performance in specific phases of the ASST-R task requires discrete regions of the prefrontal cortex and the striatum [25]. Interestingly, in an escalating model of prenatal EtOH exposure that typically results in blood alcohol concentrations of 120 mg/dL–150 mg/dL [26,27], in a separate cohort of animals [20] we observed reduced functional connectivity between the prefrontal cortex and dorsolateral striatum in adult males, while females exhibited changes in small-world organization and network integration that suggest reductions in long-range regional communication.

In the current study, we sought to determine whether the combination of postnatal choline supplementation and working memory training during adolescence might normalize functional connectivity and behavior in male and female rats exposed to EtOH gestationally. We found that as previously observed, males exposed to EtOH tended to use a non-rule-based strategy in the ASST-R task, as evidenced by a failure to form an attentional set across phases of the task. Females did form an attentional set. Prenatal EtOH increased the number of trials needed to switch from the initial dimension to the new dimension in the extradimensional shift phase in females. EtOH-exposed rats of both sexes made more errors in this task. Our intervention appeared to generate two groups in EtOH males: those that improved and exhibited a normal strategy to progress through the ASST-R task and those that did not. We compared functional connectivity between control (Con) and EtOH-exposed male and female rats, as well as between males that responded to the intervention and those that did not. This analysis revealed increased connectivity between the prefrontal cortex and striatum in those responding to treatment, as indexed by performance on the ASST-R task. Further, EtOH affected functional connectivity between the frontal cortices and the cerebellum, suggesting that disturbing cerebellar development might influence the prefrontal cortices and their modulation of circuitry involved in cognitive flexibility [28].

## 2. Materials and Methods

### 2.1. Animals

Timed pregnant Long-Evans rats were purchased from Envigo (Frederick, MD, USA). Animals were housed at the University of Maryland School of Medicine in an Association for Assessment and Accreditation of Laboratory Animal Care International-accredited facility and maintained on a 12/12 h light/dark cycle (lights on at 7 a.m.). All procedures were conducted in compliance with the guidelines for animal care established by the National Institutes of Health as well as with the approval of the Institutional Animal Care and Use Committee (protocols 1214005 and 1017010) at the University of Maryland School of Medicine.

A total of 19 dams were used in the study. Ten dams were assigned to the control (Con) group and were given ad libitum access to a liquid diet (L10251A, Research Diets, New Brunswick, NJ, USA) from gestational day (G) 6 until G 20. Nine dams assigned to the ethanol (EtOH) group were given the same liquid diet containing 3% EtOH from G 6 to G 20. Con dams received maltose-dextrin to replace calories added by ethanol in the EtOH group. Diet consumption did not differ between groups. Litters were culled to 10–12 the day after birth, keeping the same number of males and females in each litter when possible. Pups remained with the dam until postnatal day (P) 22 and were then weaned and housed with same-sex littermates in groups of 2–3.

### 2.2. Choline Administration

Pups were given subcutaneous daily injections of saline or choline chloride (100 mg/kg; Sigma-Aldrich, Burlington, MA, USA) from P 16 until P 30. Choline was administered by subcutaneous injection as previously published [21]. We chose this route of administration to control the dose given, and to avoid the stress associated with intragastric gavage and control the amount of choline administered. Pups from Con or EtOH litters were assigned to two groups: Saline Untrained (no intervention) or Choline Trained (intervention). These groups were chosen based on our published behavioral observation that working memory training coupled with choline administration improved the performance of EtOH rats on the attentional set shifting task in adulthood [21]. Neither working memory training nor choline alone were as effective as the combination of the two. Therefore, to reduce the number of comparisons on resting state functional MRI (rs-fMRI) analysis, we directly compared these two groups to determine connectivity differences between the groups. Only one male and one female from any litter was assigned to a group. The number of rats in each condition is listed in Table 1.

### 2.3. Delayed Non-Matching to Place

The delayed non-matching to place task was conducted similarly to the method described in [21]. Food restriction was initiated on P 28. On P 29, rats were placed in the T-maze with a food reward in each arm. Rats quickly began to traverse the maze and eat the reward. On P 30, training on the delayed non-matching to place task began. Each trial consisted of two phases. In phase 1, the demonstration phase, one arm of the maze was blocked and one was open. The rat was permitted to enter the open arm to retrieve the food reward. The rat was then placed in the start position for the pre-determined delay. After the delay, the barrier was removed and the rat was permitted to choose between the two open arms. The rat received a second reward if it entered the arm that was not just previously visited and reinforced. The task was conducted over eight days, with 12 trials per day. The first five sessions used a 10 s delay. The next two sessions used a 30 s delay. The final day was conducted with 0 s delay to confirm that the animals had learned the non-match rule. Untrained rats were placed in the T-maze for the same number of trials as trained rats, and both bowls contained reward. These rats experienced the T-maze but no demands were placed on working memory. All animals were returned to ad libitum access to chow on P 38. The number of correct trials and percent correct was recorded for each trained animal. Latency to reach the reward bowl was recorded for all animals.

### 2.4. Attentional Set Shifting and Reversal Learning

Attentional set shifting and reversal learning were conducted similarly to our published results [21], with the exception that for 1–2 rats per group the initial target dimension was odor, and remaining rats were trained with digging media as the target dimension. Our testing procedure was based on the work of Birrell and Brown [29]. Food restriction was initiated on P 55 and testing was conducted between P 58 and P 67. Testing was conducted in a plexiglass chamber with an open top (49 cm × 39 cm × 24 cm) under dim light. Rats progressed through a series of discriminations in which they dug in one of two small bowls for food reward. For all phases, the number of trials to criterion (the number of trials each rat required to achieve 8 correct choices in a row) was recorded. This task consisted of six phases: Phase 1 was a simple discrimination, in which only the digging media or odor cue differed between the bowls. One digging medium (e.g., large beads) or odor (e.g., garlic) predicted the presence of reward, and another (e.g., yarn or cloves) predicted the absence of reward. In phase 2, another dimension was added; for most rats, media continued to predict which bowl was baited with reward but a second dimension of odor (e.g., cinnamon or basil) was varied, and had to be learned to be irrelevant (compound discrimination). The dimensions were reversed for other rats, and odor was consistently the predictive dimension, while media was an irrelevant dimension. Rats then progressed to an intradimensional shift where the predictive dimension was the same as that used in the compound discrimination, but specific stimuli comprising the predictive and non-predictive dimensions changed. Once criterion was achieved, the contingencies were reversed. Animals then performed a second intradimensional shift, maintaining the predictive dimension used in the first 4 phases to ensure that normal rats formed an “attentional set” to the dimension [29]. An attentional set is formed when the animal uses a single aspect of a multidimensional stimulus that predicts reward. To determine whether rats formed an attentional set, an extradimensional shift, in which the predictive stimulus changed from digging media to odor or from odor to the digging media, was conducted. If rats form an attentional set, they require more trials to achieve criterion in this phase, as they require more feedback from errors to update contingencies and make this predictive shift. In normal rats, both reversal learning and performance on the extradimensional shift phase will require more trials for criterion than the phase immediately prior to each. Thus, “shift cost” (shift cost = trials to criterion in extradimensional shift-trials to criterion of the second intradimensional shift) is typically positive in normal rats. On the final day of testing, rats were returned to ad libitum access to chow for at least one week prior to undergoing rs-fMRI.

### 2.5. MRI Data Acquisition

An imaging experiment was performed on young adult offspring within P 67–74, using a 7T/30 cm Bruker Biospec horizontal bore animal scanner (Bruker Biospin MRI GmbH, Ettlingen, Germany). A Bruker 72 mm linear volume coil was used as the transmitter and a Bruker ^1^H (proton) four-channel surface coil array was used as the receiver. Anesthesia was induced with 2% isoflurane and maintained using 0.25–0.5% isoflurane in oxygen-enriched air combined with continuous infusion of dexmedetomidine (0.015 mg/kg/h) during data acquisition. A bolus dose of dexmedetomidine (0.03 mg/kg) was administered intramuscularly prior to the continuous infusion. The animal respiration rate and body temperature were monitored using an MR-compatible small animal monitoring and gating system (SA Instruments, Inc., New York, NY, USA). The body temperature was maintained at 35–37 °C using warm water bath circulation. Head motion was minimized with ear pins and bite bars.

Brain anatomical images were obtained using a two-dimensional rapid acquisition with relaxation enhancement (RARE) sequence covering the entire brain [30] (repetition time/effective echo time (repetition time (TR)/effective echo time 1(TE_eff1_)/effective echo time 2(TE_eff2_) = 3000/22/66 ms, field of view = 3.5 cm × 3.5 cm, in-plane resolution = 0.137 mm × 0.137 mm, RARE factor = 4, number of slices = 22, slice thickness = 1 mm).

The rs-fMRI was acquired by matching the anatomical images using a single-shot, gradient-echo-planar imaging sequence (TR/TE = 1000/15 ms, field of view = 3.5 cm × 3.5 cm, in-plane resolution = 0.547 mm × 0.547 mm, number of slices = 22, slice thickness = 1 mm). Six hundred volumes were taken, resulting in a total scanning time of around 10 min for each rs-fMRI run. Three or four such rs-fMRI runs were acquired on each animal.

### 2.6. Rs-fMRI Data Preprocessing

The first ten volumes were discarded from each rs-fMRI run to ensure a steady state of signal. All rs-fMRI images were then preprocessed using SPM12 (http://www.fil.ion.ucl.ac.uk/spm/) and AFNI (http://afni.nimh.nih.gov/afni). The main preprocessing steps included slice timing, realignment, normalization to a brain anatomical template [31], detrending, band-pass filtering (0.008 Hz–0.15 Hz), and smoothing (FWHM = 0.8 mm).

### 2.7. Statistical Analysis

Behavioral results of the T-maze were first analyzed using a two-way analysis of variance (ANOVA), with sex and prenatal condition as between-subject factors. The percent trial correct for each phase was the within-group variable, and experimental group (Con or EtOH) was the between-group variable. For data from the ASST-R task, three-way repeated measures ANOVA was used, with sex, prenatal treatment, and postnatal treatment (i.e., intervention or no intervention) as between-group variables. We then analyzed males and females separately to determine differences between treatment groups within sex. Previous work from our group found sex differences in both behavioral and rs-fMRI measures in our models of FASD [20,21]. Trials to criterion of each of the 6 phases was the dependent variable, and experimental group was the between-group variable.

For rs-fMRI data analysis, seed-based functional connectivity analysis was performed to investigate the effects of EtOH and the intervention (choline and working memory training) on the FASD animals. Males and females were analyzed separately. Three frontal brain regions of interest related to cognitive and executive functions were selected: prelimbic cortex (PrL), infralimbic cortex (IL), and medial-ventral orbitofrontal cortex (mvOrb). The region of interest (ROI) masks were manually delineated on the T2 weighted template of rat brain based on Paxinos and Watson [32], as shown in Figure 1. The blood oxygen level dependent (BOLD) imaging time courses for each seed were extracted as the averaged time courses from all the voxels within the region of interest. Pearson’s correlation coefficient was computed between the averaged time series of the region of interest and the time series of all the voxels within the whole brain to create the seed-based functional connectivity map. The functional connectivity map was converted to the z-score map by applying the Fisher z-transformation [33]. For the animals with four runs of rs-fMRI, the spatial correlation among the z-maps of the four runs were calculated, and the three runs that were most highly spatially correlated were selected. The averaged z-map of the three runs was calculated for each animal and subjected to a two-way ANOVA analysis (AFNI: 3d MVM) with EtOH and intervention as two factors in males and females respectively. A voxel-wise threshold of *p* < 0.005 was applied. The AFNI 3d ClustSim was used to estimate the required minimum cluster size to maintain a 5% type 1 error rate (alpha < 0.05) [33]. If a significant EtOH × intervention interaction was detected, group comparisons were performed to focus on the EtOH effect in the group without intervention (Con Saline Untrained vs. EtOH Saline Untrained) and intervention effect in EtOH group (Con Trained vs. EtOH Trained).

Based on performance in the ASST-R task, male EtOH rats receiving the intervention were divided into responders and non-responders to choline plus training. Among the 9 animals, 5 were identified as responders and 4 were identified as non-responders. Responders were defined as those rats that exhibited a shift cost of at least 2 trials to achieve the criterion of 8 consecutive trials correct, while non-responders were defined as male rats that did not exhibit an increase of trials between the second intradimensional shift and the extradimensional shift. Functional connectivity between frontal regions of interest (PrL, IL, mvOrb) and dorsal medial striatum (StrDM) were assessed and compared between responders and non-responders. The region of interest for StrDM is demonstrated in Figure 1. Pearson’s correlation was computed between the averaged time series of each pair of regions of interest (PrL-StrDM, IL-StrDM, mvOrb-StrDM). The correlation coefficient was converted to z-score using the Fisher-z transformation. Two-sample two-sided t-test was performed on the z-score with the significance threshold *p* < 0.05.

## 3. Results

### 3.1. All Groups Achieved High Response Accuracy over Training Sessions on the Delayed Non-Matching to Place Task

Repeated measures ANOVA confirmed a significant effect of session, F (6,192) = 53.68, *p* = 0.0001, as the number of correct choices increased across training sessions in both groups. A significant prenatal condition × session interaction was detected, F (6,192) = 2.57, *p* < 0.02. This interaction was driven by poor performance by Con rats in an early training session, while EtOH rats progressed quickly. This also manifested in a three way prenatal condition × sex × session interaction, F (6,192) = 3.47, *p* = 0.003, as no differences were exhibited by Con versus EtOH females. The transient poor response by Con males did not produce a significant main effect of group, F (1,32) < 1. No group differences were detected in the final test day, F (1,32) < 1. All rats performed with greater than 80% accuracy.

### 3.2. Prenatal Exposure to a Low Concentration of Ethanol Affected the Formation of an Attentional Set in a Sex-Specific Fashion

We conducted repeated measures ANOVA with trials to criterion of each of the six progressive phases of the set shifting task as the dependent variable and sex, prenatal (Con or EtOH), and postnatal treatment (choline combined with working memory training, referred to as intervention) as the independent variables (Figure 2). This analysis confirmed a significant effect of phase F (5,320) = 37.50, *p* = 0.0001, reflecting the dynamic changes in trial to criterion expected across phases. No other interactions were significant. A significant main effect of prenatal condition, F (1,64) = 10.34, *p* = 0.002 and intervention, F (1,64) = 7.75, *p* = 0.007 were confirmed. The prenatal × intervention interaction approached significance, F (1,64) = 3.07, *p* = 0.08. No interactions with sex were significant. The total number of errors made across all phases of the set shifting task were tallied (Figure 3). An effect of prenatal treatment was significant, F (1,64) = 5.71, *p* = 0.02 as was the effect of intervention, F (1,64) = 6.32, *p* = 0.01. The prenatal × intervention interaction was significant, F (1,64) = 5.05, *p* = 0.028. There was no significant interaction with sex, Fs < 1. The sexes did diverge in the final and critical phase of the attentional set shifting task. Isolation of the shift cost confirmed a main effect of sex, F (1,64) = 10.78, *p* = 0.002. The sex × prenatal treatment was significant, F (1,64) = 6.56, *p* = 0.01. Females exposed to EtOH exhibited an increase in the number of trials required to solve the extradimensional shift, while males exposed to EtOH were less likely to increase the number of trials necessary for eight correct consecutive responses relative to the immediately preceding intradimensional shift. 

### 3.3. Analysis of Males Alone Revealed a Sub-Group of Ethanol-Exposed Rats Receiving the Intervention that Employed a Normal Strategy to Solve the ASST-R Task

Repeated measures ANOVA revealed a significant effect of phase F (5,150) = 16.69, *p* = 0.0001, as the number of trials to make eight correct choices in a row changed as demands of the phase changed (Figure 2). The phase × prenatal interaction was significant, F (5,150) = 2.84, *p* = 0.017. No other interactions were significant. Further analysis determined a significant group effect on trials to criterion in the compound discrimination phase, F (3,33) = 5.43, *p* = 0.004. This was driven by poor performance in the EtOH rats without intervention. This group required more trials to criterion than the other groups, Bonferroni post hoc, all *p* values < 0.01. The main effect of postnatal treatment approached significance, F (1,30) = 3.57, *p* = 0.068, but no other main effects were significant, largest F = 2.18. The total number of errors was different between groups, F (3,33) = 2.92, *p* = 0.049 (Figure 3). Bonferroni post hoc analysis failed to find a significant difference between any pair of groups. Independent samples *t*-test found a significant difference between EtOH rats with no intervention and EtOH rats that received the intervention, *t* (16) = 0.39, *p* = 0.02, suggesting choline and working memory training reduced errors made by rats in the EtOH group. 

We categorized animals that required more than two trials to solve the extradimensional shift than the previous intradimensional shift as having made an attentional set to the dimension used in the first phases of the task (i.e., a shift cost of at least three trials; Figure 4). Those rats that did not require more than two trials were categorized as “no set formation”. Con rats were likely to make a set, with 81% requiring at least three more trials to solve the extradimensional shift while only 36% of the EtOH-exposed rats exhibited evidence of set formation, χ^2^ (1,34) = 6.27, *p* = 0.012. Chi-square analysis found a significant effect of group, χ^2^ (3,34) = 8.59, *p* = 0.035; 87.5% of Con male rats without intervention and 75% of Con male rats with the intervention formed an attentional set. Only 22% of male EtOH rats with no intervention formed a set, while 55% of male EtOH rats with the intervention formed a set. There was no significant effect on reversal cost in males, Fs < 1.2.

### 3.4. Analysis of Females Alone Confirmed Effects of Prenatal Ethanol and Intervention

Parallel analysis of females also found a significant effect of phase, F (5,170) = 22.27, *p* = 0.0001 (Figure 2). No interactions were significant. The main effect of EtOH was significant, F (1,34) = 13.57, *p* = 0.001, as was the main effect of postnatal treatment, F (1,34) = 4.29, *p* = 0.046. The interaction of EtOH and the intervention was not significant, F < 1. ANOVA examining each phase of the task found a significant effect of group on trials to criterion in the second intradimensional shift, F (3,34) = 3.05, *p* = 0.04. Bonferroni post hoc test confirmed that EtOH rats with no intervention required more trials to solve this phase than EtOH rats receiving the intervention, *p* = 0.054. There was a trend in the number of trials to criterion required to solve the extradimensional shift, F (3,34) = 2.36, *p* = 0.08. Isolation of the trials to criterion on this phase found a significant effect of EtOH, F (1,34) = 6.97, *p* = 0.012. Female rats exposed to EtOH required more trials to shift away from the originally trained attentional set. Choline combined with working memory training did not change this outcome. The total number of errors made across all phases of the task was significantly different. EtOH increased the number of errors, F (1,34) = 4.59, *p* = 0.039 (Figure 3). EtOH females receiving the intervention exhibited fewer errors than those that did not *t* (16) = 0.24, *p* = 0.03. No interactions were significant.

Analysis of reversal cost found no effect of EtOH or postnatal intervention, Fs < 1. Analysis of shift cost found a significant effect of EtOH, F (1,34) = 5.16, *p* = 0.03, as was detected in the analysis of trials to criterion. There was no interaction with the intervention. Though the intervention improved performance on the second, immediately preceding intradimensional shift, female rats exposed to EtOH required more experience with the new dimension to use it as the predictive stimulus. The majority of females formed an attentional set, regardless of experimental condition (Figure 4). In the Con groups, 85% of females formed an attentional set, while 100% in the EtOH group formed an attentional set, χ^2^ (1,35) = 2.93, *p* = 0.087.

### 3.5. Functional Connectivity of the Infralimbic Cortex and Cerebellum Was Increased in Ethanol-Exposed Rats

In male rats, significant main effect of EtOH was observed in the external cortex of the inferior colliculus and cerebellar areas (e.g., crus 1, lobule 4) *(p* < 0.005, alpha < 0.05). Con males showed significantly lower functional connectivity between IL and these regions compared to EtOH males. An EtOH × intervention interaction was observed in connectivity between IL and paramedian lobule and copula pyramidis (*p* < 0.005, alpha < 0.05). Between-group comparison showed that the functional connectivity between IL and this cluster was significantly increased in EtOH males without intervention compared to Con males without intervention (*p* < 0.005, alpha < 0.05). No significant main effect of intervention on IL functional connectivity was observed in male rats (Figure 5).

In female rats, a significant main effect of EtOH was observed in connectivity between IL and cerebellar crus 1 area (*p* < 0.005, alpha < 0.05); EtOH females had significantly higher IL functional connectivity compared to Con females. A significant main effect of intervention (*p* < 0.005, alpha < 0.05) was detected in habenular complex where intervention significantly reduced the IL functional connectivity to this area in females (Figure 6). 

### 3.6. Functional Connectivity of the Prelimbic Cortex and Cerebellum Was Increased in Ethanol-Exposed Male Rats

In male rats, a significant main effect of EtOH was observed between PrL and claustrum, caudate/putamen, the external cortex of the inferior colliculus, and cerebellar areas (e.g., crus 1, lobule 4) (*p* < 0.005, alpha < 0.05); Con males showed significantly lower functional connectivity compared to EtOH males. No significant main effect of intervention or EtOH × intervention interaction was observed on PrL functional connectivity in male rats (Figure 7). No significant main effect of EtOH or intervention was detected in female rats.

### 3.7. Functional Connectivity of the Medial-Ventral Orbitofrontal Cortex and Cerebellum Was Increased in Ethanol-Exposed Male Rats

In male rats, a significant main effect of EtOH was observed between mvOrb and ventral lateral/medial nucleus of thalamus, primary somatosensory cortex, the external cortex of the inferior colliculus, and cerebellar areas (e.g., lobule 4) (*p* < 0.005, alpha < 0.05); this was significantly lower in Con males compared to EtOH males. A significant main effect of intervention was shown in the connectivity between mvOrb and retrosplenial cortex and caudate/putamen (*p* < 0.005, alpha < 0.05); males receiving the intervention had higher functional connectivity compared to males with no intervention (Figure 8). In female rats, no significant main effect of EtOH or intervention was detected.

### 3.8. Non-Responders vs. Responders Exhibited Functional Connectivity Differences between the Frontal Cortices and the Dorsomedial Striatum

EtOH males that received the intervention were subdivided into responders and non-responders based on performance in the ASST-R task (see above) and functional connectivity between frontal regions (PrL, IL, mvOrb) and StrDM were compared between the four non-responders and five responders. Two-sample *t*-tests detected significant differences in PrL-StrDM, IL-StrDM, and mvOrb-StrDM functional connectivity (Figure 9). Responders had significantly higher PrL-StrDM (*p* = 0.035) and IL-StrDM (*p* = 0.039) functional connectivity compared to non-responders. mvOrb-StrDM functional connectivity was also higher in responders compared to non-responders (*p* = 0.051).

## 4. Discussion

The current study extends our previous results, indicating that the combination of postnatal choline supplementation and working memory training during the juvenile phase improves cognitive outcomes in rats prenatally exposed to ethanol [21]. In the prior experiment all animals started with odor as the target dimension and switched to digging media, whereas in this experiment most animals started with digging media and switched to odor. Overall, the outcomes were similar between the two studies, suggesting that this behavioral effect is not simply due to prenatal ethanol disrupting a specific sensory modality. The results presented here indicate that chronic gestational exposure to a low concentration of ethanol produces a detectable change in functional connectivity between the frontal cortices and the cerebellum (Figure 5, Figure 6, Figure 7 and Figure 8). This functional interaction is also evident during working memory engagement, and tasks that similarly engage the executive network [18,34,35]. The effects of ethanol on functional connectivity between the frontal cortices and striatum were fairly subtle at the low blood alcohol concentration in our model (estimated to result in blood alcohol concentrations of 20–30 mg/dL based on [36]), but analysis of subgroups of ethanol-exposed male rats based on the strategy used during set shifting revealed that increased connectivity was associated with normal cognitive performance (Figure 9). Postnatal choline supplementation with working memory training improved the performance of male rats on the set shifting task and increased the number of rats that formed an attentional set compared to ethanol-exposed rats with no postnatal interventions (Figure 4). Ethanol exposure did not change the likelihood that female rats would form an attentional set. Rather, females exposed to ethanol with or without the postnatal intervention appeared to require more trials during the extradimensional shift to switch from the previous predictive dimension (Figure 4).

Our behavioral intervention focuses on working memory, defined as the ability to hold information “in mind” and update contingencies in real time, because the engagement of working memory activates selective subregions of the frontal cortices [37,38]. Tests that require flexible and accurate use of working memory are the most reliable discriminative cognitive tests to distinguish alcohol-exposed versus non-exposed children [39]. Individuals that perform well on working memory tasks exhibit higher engagement of other neural substrates involved more generally in executive function, and exhibit better behavioral control than those with poor working memory performance [18,40,41]. Activation of the executive network during performance on a working memory task was similar between adults and adolescents, though activity does become more selective as individuals mature into young adulthood [18,42,43]. It is possible that choline administration might promote the generalization of working memory across cognitive domains and normalize functional connectivity between brain regions involved in executive function, as working memory training alone has been found to alter brain connectivity in healthy, typically developing humans [44,45].

Prenatal ethanol increased the connectivity between the frontal cortices and the right posterior cerebellum (Figure 5). The cerebellum appears to be involved in holding information during the delay in working memory tasks [46,47]. Interestingly, cerebellum volume correlates with working memory performance and set shifting performance [48]. Imaging data collected across a wide developmental span found that gray matter volume in the right posterior cerebellum positively correlated with working memory task performance in humans (specifically CrusI/II/VIIB) [48,49]. Increased connectivity between frontal and parietal cortices and cerebellum was evident in children with FASD compared to non-ethanol-exposed controls [50]. This increased cerebellar connectivity was not detected to the same degree in children with heavy alcohol exposure, suggesting this compensatory shift in connectivity was abolished in this group [50]. Increased cerebellar connectivity in children with FASD was detected even when performance did not differ from normal controls, when the task demands or working memory load were low [51]. This pattern of results suggests that the concentration of ethanol used in the current study parallels this compensatory change observed in humans. The recent resurgence of research into the cerebellum’s role in cognition (as opposed to movement/coordination) suggests an organizational role of the developing cerebellum on neocortical organization (reviewed in [28]). By extension, disturbed cerebellar development leads to disturbed neocortical development in distant circuits [28]. Perinatal ethanol exposure is known to disrupt circuitry within the cerebellum. Pavlovian eyeblink conditioning with parameters requiring plasticity of the cerebellar cortex and deep cerebellar nuclei is profoundly slowed in both rats and humans exposed gestationally to ethanol [52,53,54].

Choline supplementation can have robust effects on brain development and function through its three known mechanisms: (1) as phosphatidylcholine, a component of lipid membranes; (2) as a precursor to the neurotransmitter acetylcholine; and (3) as a methyl donor, modulating gene expression. Most relevant to the results presented here are choline’s enduring effects on acetylcholine transmission and DNA and histone methylation. Dietary choline can increase acetylcholine release when cholinergic neurons are activated (reviewed in [55]). Cholinergic systems modulate the development of brain regions and connectivity between brain regions involved in learning, memory, and executive function [56]. Choline supplementation during gestational and early postnatal development can improve performance in a broad range of cognitive domains throughout the lifespan, including working memory, reference memory, problem solving, and interval timing [10,11,12,57]. Choline is also oxidized to betaine, and through a methionine intermediate, donates a methyl group in a metabolic pathway that produces S-adenosylmethionine (reviewed in [58,59,60]). The majority of methylating enzymes use methyl groups from S-adenosylmethionine, making choline one of many micronutrients that contribute to DNA and histone methylation to influence gene expression (reviewed in [58]). The manipulation of choline levels in maternal diet can change methylation patterns in offspring [59,61,62].

One of the most consistent findings across rodent models with various means of delivery, concentrations, and timing of prenatal alcohol exposure is poor executive function and cognitive flexibility. The neurological underpinnings of cognitive deficits point to poor prefrontal cortex organization, and dysregulated functional cooperation between the frontal cortices and striatum (e.g., [17,19,20]). This sensitivity to ethanol exposure persists from the prenatal phase to adolescence [19,63]. Dysregulated frontal and striatal interaction is apparent in poor reversal learning ability in rodents exposed to prenatal ethanol [19,64]. Cholinergic activity in the dorsomedial striatum is critically involved in reversal learning (reviewed in [65]). Acetylcholine release increases during reversal learning specifically, and not during initial learning (e.g., [66]). Choline similarly increases in the human striatum during reversal learning, perhaps indicating an increase in acetylcholine efflux [67]. The striatum exhibits learning-related changes as learning and mastery of spatial working memory tasks similar to the one used here progresses [68]. The dorsomedial striatum exhibits high activity during early learning, which requires focus and attention; once accurate performance is achieved, the dorsolateral striatum increases activity, supporting the execution of procedural learning [68]. This pattern of results suggests the dorsomedial striatum is engaged during early working memory training—a phase of learning also likely facilitated by cholinergic transmission.

The attentional set shifting task requires the rat to attend to one dimension of a complex stimulus arrangement to gain a reward. When the predictive dimension changes, more experience is necessary to collect the information necessary to learn the new contingency. The rodent medial prefrontal cortex (mPFC) is necessary for efficient predictive dimension switches. Rats with mPFC lesions require extra trials to attend accurately to the new predictive dimension [25,29]. Rats with lesions of the orbitofrontal cortex do not form an attentional set; the number of trials between the extradimensional shift and the preceding intradimensional shift is the same [69,70]. This dissociation between subregions of the frontal cortex is also true for reversal learning. Lesions of the mPFC do not affect reversal learning, while orbitofrontal lesions increase the number of perseverative errors [70,71,72]. Lesions of the dorsomedial striatum parallel those of orbitofrontal lesions, increasing errors in reversal learning and abolishing the formation of an attentional set [73]. These results suggest that connectivity between the orbitofrontal and dorsomedial striatum specifically are involved in attentional set formation. Our behavioral results were sexually dimorphic, with females exposed to ethanol requiring more trials to switch from the previous to the current predictive dimension used in the extradimensional shift. This suggests poor mPFC function based on lesion studies (e.g., [29]). We detected increased connectivity between the infralimbic subregion of the mPFC and cerebellum in ethanol-exposed females. A similar finding in males was not associated with increased trials during the extradimensional shift in those exposed to ethanol. Dysregulated connectivity of the orbitofrontal cortex was detected in males, perhaps contributing to their use of a different strategy during the set shifting task.

We did not manipulate hormonal status during sex-specific perinatal organization of the brain, or the activational effects of hormones in later development, so the potential role of hormones in the sex differences reported here is unknown. Across many studies of male and female rodents, prenatal ethanol affects males and females differently (e.g., [74,75,76,77,78,79]). The pattern of our results is quite similar to those of Rodriguez et al. [80]. Analysis of functional connectivity of male and females gestationally exposed to moderate levels of ethanol found increased connectivity of the cerebellum with cortical areas in males, but only the striatal area in females [80]. Thus, increased cerebellar connectivity might reflect sex-specific differences in brain organization in response to EtOH. Cerebellar hyperconnectivity might disrupt the development of normal functional connectivity and differentiation of subregions of the frontal cortex observed in the typically developing brain [17]. Though the cerebellum is typically viewed as vulnerable to injury by EtOH when given postnatally in rodent models, it is possible that gestational EtOH exposure reduces the functional differentiation of the frontal cortices by increasing connectivity to the cerebellum. The contribution of cerebellar dysfunction might only be detected when the frontal cortices are also engaged. The idea that the cerebellum contributes to cortical organization has been proposed in other disorders of brain development, such as autism, which is characterized by male vulnerability [28]. The response to choline might also be sexually dimorphic. Choline given concurrently with EtOH to pregnant mice changed neocortical connectivity, making intraconnectivity more like controls, and changed behavior in a sex-specific manner [81]. In our study, choline administration did not begin until postnatal day 16, and working memory training began in the juvenile phase. It is interesting that choline influenced connectivity at both developmental phases. Further experimentation is needed to understand sex differences in sensitivity to ethanol and its effects on executive function. It is noted that children of both sexes are affected by alcohol exposure in utero, though females are less likely to manifest overt neuroanatomical changes compared to males [82,83].

## 5. Conclusions

The data presented here recapitulate functional connectivity changes observed in children exposed to ethanol and support the idea that combining nutritional support and cognitive interventions might improve cognitive flexibility in the long term. Our results suggest that interventions in a developmental phase akin to childhood can be of benefit, and influence brain connectivity. Further experiments are necessary to determine the efficacy of these interventions in higher concentrations of ethanol exposure, and interactions with biological sex, to investigate the mechanism(s) by which the interventions work, and to understand what predicts whether an intervention will be successful.

## Figures and Tables

**Figure 1 nutrients-12-03513-f001:**
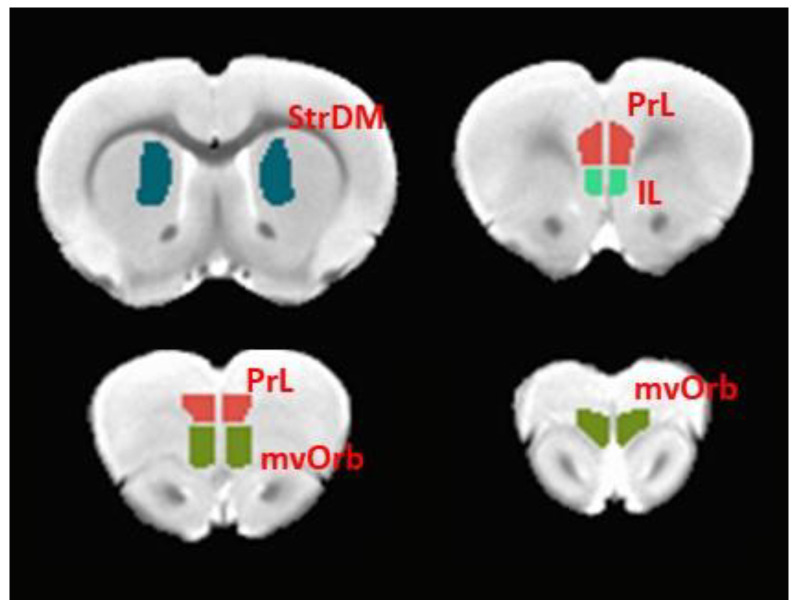
Demonstration of regions of interest for resting state functional MRI (rs-fMRI) data analysis. IL: infralimbic cortex; PrL: prelimbic cortex; mvOrb: medial-ventral orbitofrontal cortex; StrDM: dorsomedial striatum.

**Figure 2 nutrients-12-03513-f002:**
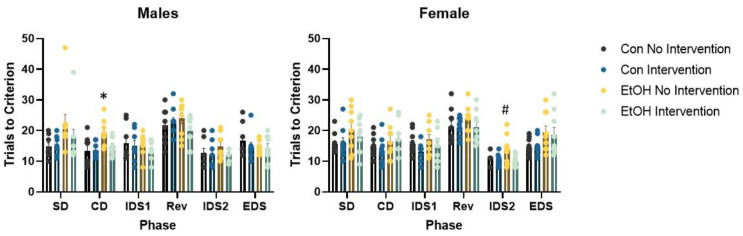
Attentional set shifting and reversal learning in adult rats. Prenatal ethanol increased the trials to criterion in some phases of the task in both males and females. The intervention, choline administration combined with working memory training, improved performance in both sexes. Analysis of males confirmed a significant phase x prenatal interaction, *p* = 0.017, and a significant increase in trials to criterion in the EtOH No intervention group on the compound discrimination phase (CD), Bonferroni posthoc, * *p* = 0.01. Analysis of females confirmed a significant effect of prenatal ethanol, *p* = 0.001, and postnatal treatment, *p* = 0.046. A significant increase in trials to criterion in the EtOH No Intervention group on the second intradimensional shift (IDS2), Bonferroni posthoc, ^#^
*p* = 0.054. Con = Control, EtOH = Ethanol, SD = Simple Discrimination, CD = Compound Discrimination, IDS1 = Intradimensional Shift 1, Rev = Reversal, IDS2 = Intradimensional Shift 2, EDS = Extradimensional Shift.

**Figure 3 nutrients-12-03513-f003:**
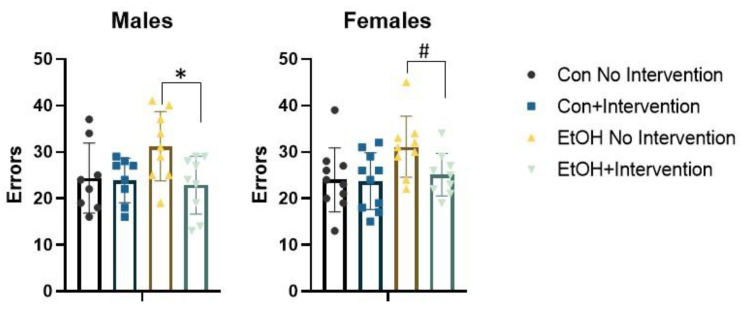
The total number of errors made during the attentional set shifting and reversal learning task in adult rats. Prenatal ethanol increased the number errors in males, *p* = 0.049, and females, *p* = 0.039. The intervention reduced errors in EtOH rats in males, * *p* = 0.02, and females ^#^
*p* = 0.03. Con=Control, EtOH=Ethanol.

**Figure 4 nutrients-12-03513-f004:**
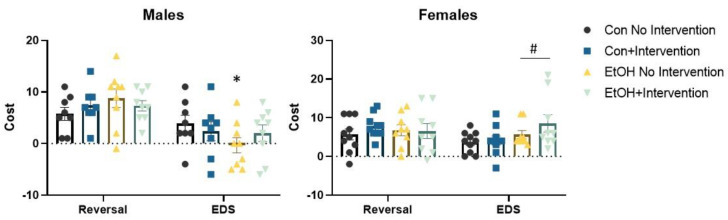
The reversal and shift cost for male and female rats. No significant effects on the reversal cost were detected. Analysis of the cost of the extradimensional shift (EDS) confirmed a main effect of sex, *p* = 0.002 and sex x prenatal treatment interaction, *p* = 0.01. Prenatal ethanol exposed male rats were less likely exhibit a positive shift cost, chi square analysis, * all *p* values < 0.035. The intervention significantly increased the number of rats exhibiting a shift cost. Prenatal ethanol did not change the number of females at exhibited an increased shift cost, but the cost was increased in females, regardless of the intervention, ^#^
*p* = 0.03. Con = Control, EtOH = Ethanol, EDS = Extradimensional Shift.

**Figure 5 nutrients-12-03513-f005:**
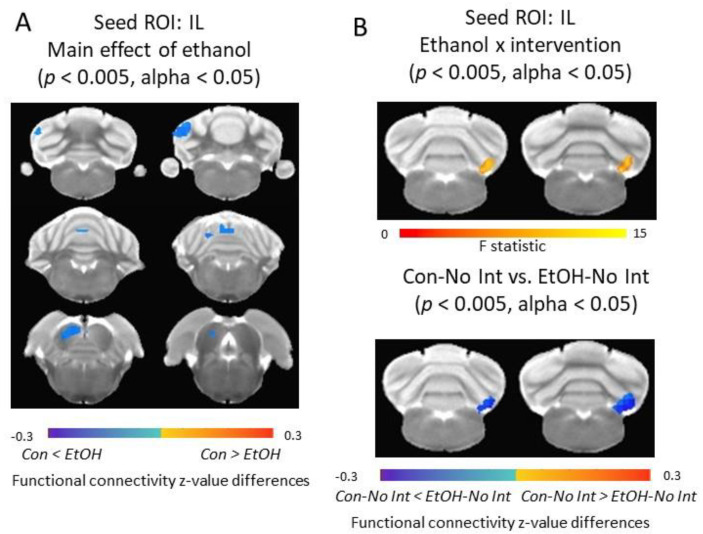
Alterations of IL functional connectivity in male rats. (**A**) Main effect of ethanol was detected in the external cortex of the inferior colliculus and cerebellar areas. Color code: connectivity differences between Con and EtOH animals. (**B**) The ethanol x intervention effect was detected in the paramedian lobule and copula pyramidis area. Color code: top panel—F statistics of the two-way ANOVA analysis; bottom panel—connectivity differences between Con-No Int and EtOH-No Int animals. ROI = Region of Interest; IL = infralimbic cortex; Con = Control, EtOH = Ethanol, No Int = No Intervention.

**Figure 6 nutrients-12-03513-f006:**
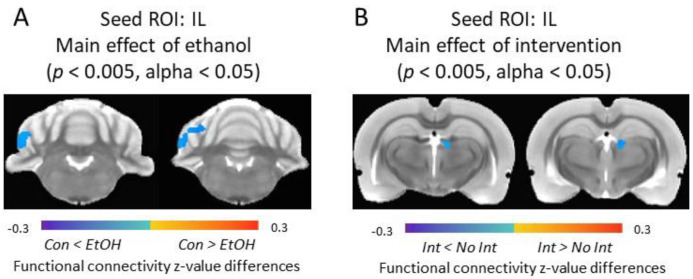
Alterations of IL functional connectivity in female rats. (**A**) Main effect of ethanol was detected in cerebellar areas. Color code: connectivity differences between Con and EtOH animals. (**B**) Main effect of intervention was detected in the habenular complex area. Color code: connectivity differences between receiving the intervention (Int) and those that did not (No Int). ROI = Region of Interest, IL = Infralimbic Cortex, Con = Control, EtOH = Ethanol.

**Figure 7 nutrients-12-03513-f007:**
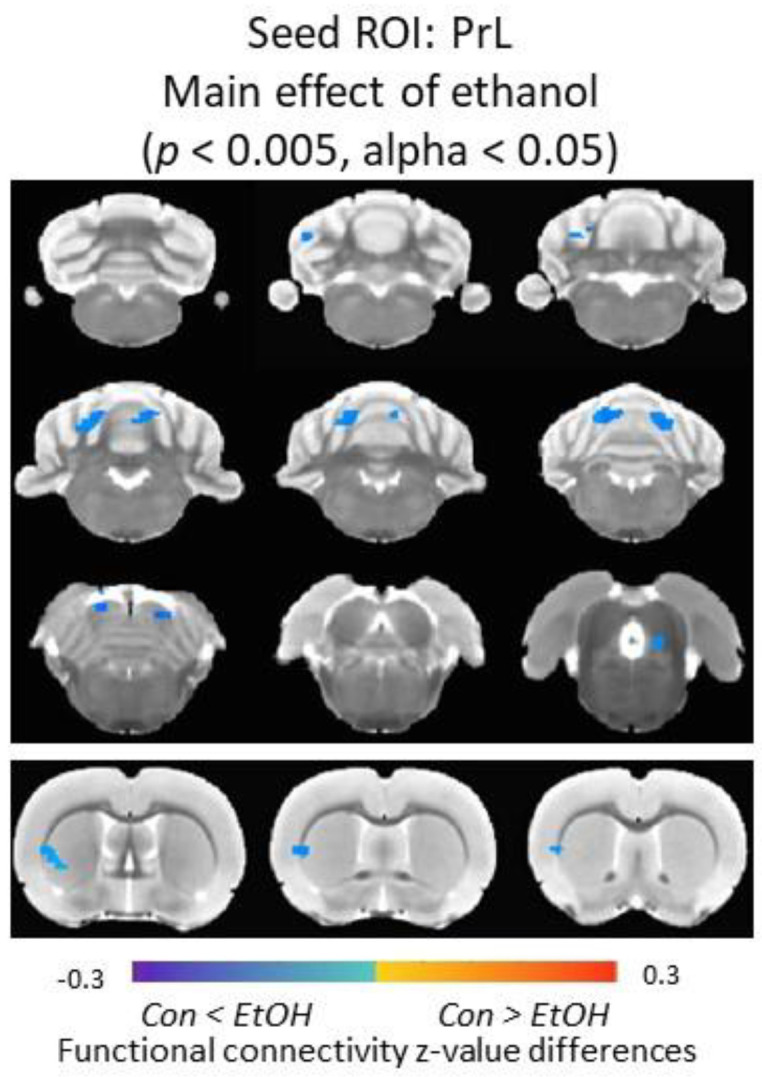
Alterations of PrL functional connectivity in male rats. Main effect of ethanol was detected in insular/striatum area, the external cortex of the inferior colliculus and cerebellar areas. Color code: connectivity differences between Control (Con) and Ethanol (EtOH) animals. ROI = Region of Interest, PrL = Prelimbic Cortex.

**Figure 8 nutrients-12-03513-f008:**
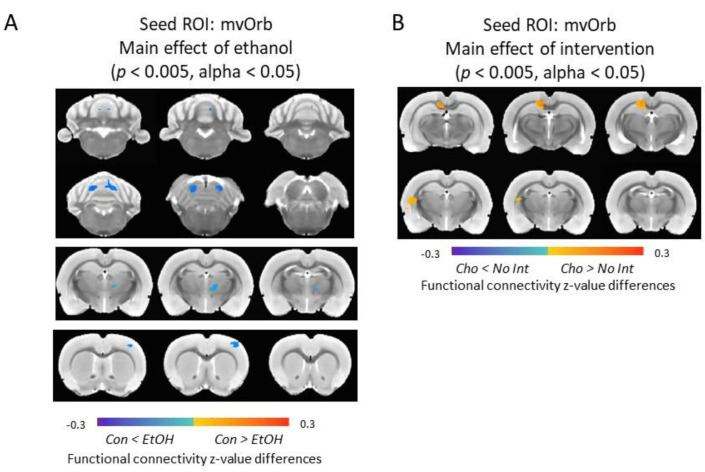
Alterations of mvOrb functional connectivity in male rats. (**A**) Main effect of EtOH was detected in ventral lateral/medial nucleus of thalamus, primary somatosensory cortex, the external cortex of the inferior colliculus and cerebellar areas. (**B**) Main effect of intervention was detected in the retrosplenial cortex and caudate putamen. Color code: connectivity differences between Control (Con) and Ethanol (EtOH) animals. ROI = Region of Interest, mvOrb = medial-ventral Orbitofrontal Cortex.

**Figure 9 nutrients-12-03513-f009:**
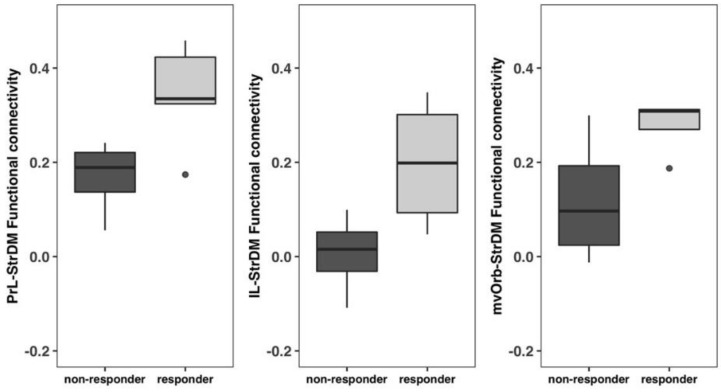
Functional connectivity differences between responders and non-responders in ethanol exposed male rats. Responders had significantly higher Prelimbic Cortex to dorsomedial Striatum (PrL-StrDM) and infralimbic to dorsomedial Striatum (IL-StrDM) functional connectivity compared to non-responders. Increased medial-ventral Orbitofronal cortext-dorsomedial Striatum (mvOrb-StrDM) functional connectivity in responders compared to non-responders was also detected. The dots represent animals outside 1.5 times the interquartile range.

**Table 1 nutrients-12-03513-t001:** Number of rats in each experimental group.

Sex	Control		Ethanol	
	Choline Trained	Saline Untrained	Choline Trained	Saline Untrained
Male	8	8	9	9
Female	10	10	9	9

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
