# Peer review of "Choline Plus Working Memory Training Improves Prenatal Alcohol-Induced Deficits in Cognitive Flexibility and Functional Connectivity in Adulthood in Rats"

_nutrients, 2020, doi:10.3390/nu12113513_

Round 1

Reviewer 1 Report

In this manuscript, Waddell and colleagues investigated if the combination of Choline supplementation and working memory training during adolescence might recue the behavioral deficits and the functional connectivity in male and female rats exposed to ethanol gestationally. They found that the rodents exposed to ET gestationally show functional connectivity deficits similar to the changes observed in children exposed to ethanol. Further combining ethanol and working memory training during adolescent period might improve cognitive flexibility long term.

In general, this was a well conducted and described series of experiments. The results could have a high impact on the field, given the limited treatment option for FASD. However, there are several issues with the methodology of the ASST that need to be addressed.  In addition, a more thorough description of the fMRI methods is required to help non-experts understand the impact of the results.

Major Comments:

  1. Only previous data was cited for estimating BAC of the animals. Were BAC or drinking levels calculated in the current study? Please comment on how you would account for the amount of alcohol consumed?
  2. The term “adolescent” and “juvenile” are used interchangeably between this and previous studies which is confusing since it appears to be different periods although the intervention occurs between P16-P-30 in both studies. Please be consistent with terminology
  3. Please describe the mating period - was there a fixed time, number of days?
  4. Was more than 2 offspring per litter was used? What methods were used to statistically account for litter effects?
  5. Why were there only 1-2 groups trained on odor as the initial dimension? It is well known that stimulus dimensions in the ASST are not equally salient during early learning. Odor dimensions are typically much easier to learn than tactile. Please more clearly describe why this approach was taken and report if there was an effect of dimension during learning.
  6. Weight reduction- line 143 ‘Food restriction was initiated on P55 and testing was conducted between P58 and P67’ Why was the training started immediately following the start of food restriction? Is there a reason for not waiting for weight reduction before start of training? How much food restriction was done? Was there a target weight?
  7. Rats were not housed in reverse dark cycle. Were the experiments performed during the light phase? If so, will there be behavioral effects of changing to dim light during the light phase? Please justify testing nocturnal animals during the light phase.
  1. Regarding statistical analysis, performing t-test between just two groups and stating the results when there are more than 2 groups is not appropriate. Was this an a priori comparison that was conducted? If so, this needs to be stated in the methods or intro.
  2. Regarding the fMRI figures- It would be easier to understand if the fmri images are clearly labelled (CT, ET) to aid in interpretation.

Minor comments:

Results Line 263 pg 6 – The figure number is referenced incorrectly (its figure 3 and not figure 4)

Reviewer 2 Report

Review of Waddell et al. for Nutrients.  October 13,2020

Title: Choline Plus Working Memory Training Improves Prenatal Alcohol-Induced Deficits in Cognitive Flexibility and Functional Connectivity in Adulthood in Rats.

FASD, caused by prenatal ethanol exposure through maternal consumption during pregnancy, is a spectrum disorder that affects a great deal of children in the US.  Many FASD researchers have detailed the effects of prenatal ethanol exposure in rodent systems, and some have investigated potential mechanisms that contribute to the phenotype.  Also, more recently, choline, an essential nutrient, has shown to play an ameliorative effect on the development of FASD when rodent dams are administered choline along with ethanol.  While this preventative effect of choline is promising, it is extremely important to discover whether choline supplementation after offspring are born with prenatal ethanol exposure-induced phenotypes.  In this report, the authors’ describe an insightful series of experiments that test how choline administration with specific training, during a rodent age period equivalent to human adolescence, may be of therapeutic value.

INTRODUCTION:  This is a curt and concise description of background information relevant to the study. The authors do an excellent job is explaining why a study in working memory is important in a rat model of FASD, and how the data can relate to human data showing deficits in working memory from prenatal ethanol exposure.

METHODS: The design is very good, however, given that the ethanol was self-administered in the bottle, it is puzzling why the authors did not also administer the choline by mouth.  Although saline injection controls were used, it is possible that choline may mediate pain or inflammatory responses after the injection. It is an unlikely, but possible confound.  The methods for DNMP and attentional set-shifting and reversal learning were described well and conducted properly. Minor detail: The use of CT and ET are confusing. I first assumed this meant Choline Treated and Ethanol Treated.  I then realized CT was control.  Perhaps changing it to CON and EtOH may help prevent confusion.

RESULTS: Although power may be low, the behavioral results would be strengthened by also conducting a 3-way ANOVA where sex is a variable, rather than running two-ways independently for males and females.  There are many analyses for the various behavioral measures and the MRI data, I think it would greatly benefit the reader if the authors created a results summary table, stating main and interaction effects for males and females on the multiple measures.  There are many important significant findings in the manuscript and figures, especially with the low dose paradigm, so a summery table would be helpful to make it very clear.  This is especially important as the effects clearly vary tremendously by sex.  Also, on figures 2-4, there are no indications of significant group differences (like by asterisk). The legends mention group differences but not p values or statistical significance.

DISCUSSION: While the discussion is well written, and connects the finding to the lab’s past research and established links between the brain and behavioral data, it lacks a discussion of the sex differences.  The male and female differences, in response to the intervention, warrant some discussion about what may account for those differences.  In a recent paper from Bottom et al. (2020) in Neuropharmacology, responses to choline supplemental varied significantly by sex; thus, some hypotheses about why choline may have differential effects on males and female would be prudent.  This recent report should be added to the discussion as it documents frontal connectivity issues in ET mice and the partial rescue of these phenotypes with choline administration.

Overall, this report reflects valuable findings with high impact results. 

Reviewer 3 Report

This paper uses functional mRI to explore the effects of an intervention (choline plus memory training) on the cognitive flexibility and brain connectivity in rats following prenatal exposure to alcohol. The paper describes some complex data but is very clearly written and the experiments are well designed and justified.

The strengths of the study include the use of a relatively low dose of alcohol which mimics the most common form of alcohol consumption by women, thus increasing the clinical relevance of the study. Also, the authors have pleasingly shown all data points on their graphs allowing the reader to clearly see the variance amongst the groups.

Of great interest is the sex differences: this is discussed briefly by the authors in the discussion (lines 491-500) but without any real insights into what may be the underlying mechanisms. As the authors highlight, children of both sexes are affected by prenatal alcohol exposure and if interventions are to be effective, we need to understand if they will only work in one sex.

Minor points:

Methods lines 199-120- it is unclear which are the two groups referred to in line 120. Is this the control and the combination (of choline and memory training)?

Line 200 – sentence needs to be reworded: Males and females were analyzed separately to retain sufficient power

Line 227 – 232. This refers to males either responding or not. Did all females ‘respond’? Although this might be implicit, it would help the read to have this stated explicitly.

Results: In the results, the p value is often very specific (eg line 262 p<0.068 and line 263, P<0.049). In this case, I don’t think it appropriate to use the < sign, this should be given as p=0.068 or p=0.049. The < sign could be used if you were using the convention of P<0.05, P<0.01 and P<0.001 (as is used later in the paper when discussing the rsfMRI results.

Author Response

Please see the attachment. Please note this is the response to the Editor's comments.
